# Assessing Fire Risk in Wildland–Urban Interface Regions Using a Machine Learning Method and GIS data: The Example of Istanbul's European Side

Ercüment Aksoy [1], Abdulkadir Kocer [2], İsmail Yilmaz [3], Arif Nihat Akçal [4] and Kudret Akpinar [5,*]

1   Department of Geographic Information Systems, Akdeniz University, Antalya 07070, Türkiye; ercumentaksoy@akdeniz.edu.tr
2   Department of Electricity and Energy, Akdeniz University, Antalya 07070, Türkiye; akocer@akdeniz.edu.tr
3   Remote Sensing and Geographic Information Systems Graduate Program, Institute of Natural and Applied Sciences, Akdeniz University, Antalya 07070, Türkiye; isoylmz@gmail.com
4   Department of Construction, Akdeniz University, Antalya 07070, Türkiye; arifnihatakcal@akdeniz.edu.tr
5   Department of Civil Defense and Firefighting, Akdeniz University, Antalya 07070, Türkiye
*   Correspondence: akudret@akdeniz.edu.tr; Tel.:+90-242-3106741

**Abstract:** Like many places around the world, the wildland–urban interface areas surrounding urban regions are subject to variable levels of fire risk, threatening the natural habitats they contact. This risk has been assessed by various authors using many different methods and numerical models. Among these approaches, machine learning models have been successfully applied to determine the weights of criteria in risk assessment and risk prediction studies. In Istanbul, data have been collected for areas that are yet to be urbanized but are foreseen to be at risk using geographic information systems (GIS) and remote sensing technologies based on fires that occurred between 2000 and 2021. Here, the land use/land cover (LULC) characteristics of the region were examined, and machine learning techniques, including random forest (RF), extreme gradient boosting (XGB), and light gradient boosting (LGB) models, were applied to classify the factors that affect fires. The RF model yielded the best results, with an accuracy of 0.70, an F1 score of 0.71, and an area under the curve (AUC) value of 0.76. In the RF model, the grouping between factors that initiate fires and factors that influence the spread of fires was distinct, and this distinction was also somewhat observable in the other two models. Risk scores were generated through the multiplication of the variable importance values of the factors and their respective layer values, culminating in a risk map for the region. The distribution of risk is in alignment with the number of fires that have previously occurred, and the risk in wildland–urban interface areas was found to be significantly higher than the risk in wildland areas alone.

**Keywords:** fire risk; wildland–urban interface; rural–urban interface; machine learning classification; GIS

## 1. Introduction

It has been widely reported that rural areas and forests are subject to significant encroachment and pressure due to rapid population growth [1–3]. This pressure has led to an increase in forest fires and fires in rural areas. Despite the developments in recent years concerning firefighting technologies, increased efficiency in fire prevention activities, reforestation and improvement efforts, and heightened awareness of fires, a report by the Food and Agriculture Organization of the United Nations concluded that the reduction of forests has not been mitigated [4]. These total global forest losses are partly due to fires and partly due to losses in forest quality because of land use. However, regardless of the reasons, the decrease in forests and vegetation means less absorption of sunlight and atmospheric carbon, which will clearly contribute to global warming. The increased global warming brings more drought—particularly in the areas in the Northern Hemisphere that are close to the equator—due to climate change and causes conditions of low humidity, which results

in forests being more susceptible to fire. Once a fire starts under such conditions, it spreads faster and becomes more difficult to extinguish.

The point to be emphasized is not the policy of immediate response to all kinds of forest fires that was once implemented but the increase in the frequency and severity of forest fires in recent years with global warming. In the wildland–urban interface areas in the region that is the subject of this study, problems related to the losses of property and lives and the increases in firefighting budgets are especially important [5,6]. Such problems increase the importance of risk assessment studies in terms of taking precautions against fires, planning, and early detection.

The primary cause of fires is predominantly human activities. In fires that occur in rural and forested areas, natural causes play a minimal role in ignition and spread, whereas direct or indirect anthropogenic factors are significantly more prevalent [7,8]. In the United States, it is estimated that the 12% increase in fires ignited by lightning strikes is due to increased global warming [9]. Even with this increase, human-induced fires still make up a number far below the estimated 80% of all fires [10].

In this context, areas where rural and forested lands are intermixed with human settlements are increasingly influenced by human activities. If human settlements are in close proximity to natural and anthropogenic biomass, the primary cause of forest, rural, and biomass fires is predominantly human activities. In their review of definitions and approaches in the literature, Lampin-Maillet et al. (2010) noted that areas characterized by increased human activities and transformations in land use are termed wildland–urban interfaces (WUIs). They also emphasized that the importance of WUIs has increased globally, particularly in recent years, as they are considered as landscape units [11]. In fires that occur in WUIs, the lives and properties of residents are also at risk [12]. To minimize these dual risks, it is crucial that urban planners and relevant agencies take the findings of fire-related studies into consideration. As cities expand into natural areas, they also undergo various on-site transformations to secure needed resources, such as water, energy, transportation, etc. Examples include constructing roads, extending electrical lines, building dams, laying pipelines, and opening plantation areas.

Bento-Gonçalves and Vieira (2020) reported that, in addition to areas where urban lands meet natural areas, which have an interface characteristic, some authors in fire-related topics have introduced different terms, such as rural–urban interface (RUI) and urban–brushland interface (UBI), for places where agricultural areas, rocky terrains, shrubs, or reed beds intersect or blend with urban boundaries. However, they also noted that the term wildland–urban interface (WUI) is broadly used not just for forests, but also for the areas mentioned above.

In studies of fires in WUI areas, some factors that are commonly considered for fires that occur in wildlands and forests are the distance to roads, vegetation, slope and position of houses on hills, construction materials used in homes, fire and support structures, and urban infrastructures, particularly power lines [13]. In the assessment of risks in rural, forest, and vegetation fires, the criteria generally selected include land slope, distance to roads, elevation, distance to water bodies, aspect, types of vegetation, land use, population, climate features, distance to settlements, and types of fuel [14–17]. Factors such as slope, elevation, aspect, distance to water bodies, and climate features are directly related to geography. On the other hand, factors such as population, distance to settlements, distance to roads, and land use are directly human-induced, and factors such as vegetation, type of fuel, and even climate features can be indirectly related to human activities. Vegetation not only exists through natural growth, but may also be destroyed or modified in many areas by human activities, such as agriculture and forestry, and, thereby, exists under anthropogenic domination. Thus, humans partially influence what types of fuel for fires exist in rural and forested areas.

The factors commonly used for fire risk assessment in forest and rural fires are provided in Table 1 based on a review of modeling studies [14–39].

**Table 1.** Factors used in fire risk assessment.

| Factors | [33] | [24] | [27] | [36] | [30] | [39] | [29] | [31] | [14] | [21] | [20] | [25] | [15] | [26] | [37] | [16] | [17] | [23] | [19] | [38] | [35] | [18] | [34] | [32] | [22] | [28] |
|---|---|---|---|---|---|---|---|---|---|---|---|---|---|---|---|---|---|---|---|---|---|---|---|---|---|---|
| Slope | x | x | x | x | x | x | x | x | x | x | x | x | x | x | x | x | x | x | x | x | x | x | x | x | x | x |
| Aspect | x |  |  | x |  | x | x | x | x | x | x | x | x | x | x | x | x | x | x | x | x |  | x | x | x | x |
| Elevation | x |  | x | x | x | x |  | x | x | x |  | x | x | x | x | x | x |  |  | x |  |  | x | x | x | x |
| Distance to settlements | x | x |  |  | x | x | x | x | x | x | x | x | x |  | x | x | x | x |  | x |  |  | x | x | x | x |
| Distance to roads | x | x | x |  | x | x | x | x |  | x |  | x | x |  | x | x | x | x |  | x |  |  | x | x | x |  |
| Distance to water bodies |  |  | x | x |  |  |  | x | x |  | x |  |  |  | x |  |  | x |  |  |  |  |  | x | x |  |
| Land use |  |  |  | x | x |  | x | x |  |  |  |  |  | x |  |  |  |  |  | x |  |  | x | x |  | x |
| Precipitation |  |  | x |  | x |  |  |  |  |  | x |  | x | x |  |  |  |  |  | x | x |  |  |  |  | x |
| Vegetation density |  |  |  |  |  |  |  |  |  | x | x |  |  |  |  | x | x |  |  |  | x |  |  |  |  |  |
| Temperature |  |  | x |  |  |  |  |  |  |  |  |  |  | x | x |  |  |  |  | x |  |  |  |  |  | x |
| Plant type |  | x |  |  |  | x |  |  |  | x |  |  |  |  |  |  |  |  |  |  |  |  | x | x |  |  |
| Distance from agricultural land |  |  | x |  |  |  |  |  |  | x |  |  |  |  | x | x |  |  |  |  |  |  |  |  |  |  |
| Wind speed |  |  |  |  |  |  |  |  |  |  | x |  | x | x |  |  |  |  |  |  |  |  |  |  |  | x |
| Stand crown closure |  |  |  |  |  |  |  |  |  | x | x |  |  | x |  |  |  |  | x |  |  |  |  |  |  |  |
| Population |  |  |  |  |  |  |  |  |  | x |  |  |  |  | x |  |  |  |  |  |  |  |  |  |  |  |
| Topographic Wetness Index |  |  |  |  |  |  | x |  |  |  |  | x |  |  | x |  |  |  |  |  |  |  |  |  |  |  |
| Canadian Forest Fire Weather Index (FWI) |  |  |  |  |  |  |  |  |  | x |  |  |  |  |  |  | x |  |  |  |  |  |  |  |  |  |
| Tree stage |  |  |  |  |  |  |  |  |  |  | x |  |  |  |  |  |  |  |  |  | x |  |  |  |  |  |
| Fuel type |  |  |  |  |  |  |  |  |  |  |  |  | x |  |  |  | x |  |  |  |  |  |  |  |  |  |
| Humidity |  |  |  |  |  |  |  |  | x |  |  |  |  |  |  |  |  |  |  |  |  |  |  |  |  |  |
| Forest type | x |  |  |  |  |  |  |  |  |  |  |  |  |  |  |  |  |  |  |  |  |  |  |  |  | x |
| Distance to tourist places |  |  |  |  |  |  | x |  |  |  |  |  |  |  |  |  |  |  |  |  |  |  |  |  |  |  |
| Distance from an anti-poaching camp shed |  |  |  |  |  |  | x |  |  |  |  |  |  |  |  |  |  |  |  |  |  |  |  |  |  |  |
| Distance to fields |  |  |  |  |  |  |  |  | x |  |  |  |  |  |  |  |  |  |  |  |  |  |  |  |  |  |
| Forest cover |  |  |  |  |  |  |  |  | x |  |  |  |  |  |  |  |  |  |  |  |  |  |  |  |  |  |
| Distance to previous fire points |  |  |  |  |  |  |  |  |  | x |  |  |  |  |  |  |  |  |  |  |  |  |  |  |  |  |
| Tree species |  |  |  |  |  |  |  |  |  |  | x |  |  |  |  |  |  |  |  |  |  |  |  |  |  |  |
| Topographic Position Index (TPI) |  |  |  |  |  |  |  |  |  |  |  | x |  |  |  |  |  |  |  |  |  |  |  |  |  |  |
| Land surface temperature |  |  |  |  |  |  |  |  |  |  |  | x |  |  |  |  |  |  |  |  |  |  |  |  |  |  |
| Bare soil index |  |  |  |  |  |  |  |  |  |  |  |  | x |  |  |  |  |  |  |  |  |  |  |  |  |  |
| Species composition |  |  |  |  |  |  |  |  |  |  |  |  |  | x |  |  |  |  |  |  |  |  |  |  |  |  |
| Development stage |  |  |  |  |  |  |  |  |  |  |  |  |  |  | x |  |  |  |  |  |  |  |  |  |  |  |
| Solar radiation |  |  |  |  |  |  |  |  |  |  |  |  |  |  | x |  |  |  |  |  |  |  |  |  |  |  |
| Fire regime (TSF-FR) |  |  |  |  |  |  |  |  |  |  |  |  |  |  |  |  | x |  |  |  |  |  |  |  |  |  |
| Tree species composition |  |  |  |  |  |  |  |  |  |  |  |  |  |  | x |  |  |  |  |  | x |  |  |  |  |  |
| Topomorphology |  |  |  |  |  |  |  |  |  |  |  |  |  |  |  |  |  |  |  |  |  |  |  | x |  |  |
| Soil use |  |  |  |  |  |  |  |  |  |  |  |  |  |  |  |  |  |  |  |  |  |  |  | x |  |  |
| Distance to fire response teams |  |  |  |  |  |  |  |  |  |  |  |  |  |  |  |  |  |  |  |  |  |  |  |  | x |  |
| Distance to fire watch towers |  |  |  |  |  |  |  |  |  |  |  |  |  |  |  |  |  |  |  |  |  |  |  |  | x |  |
| Visibility from fire watch towers |  |  |  |  |  |  |  |  |  |  |  |  |  |  |  |  |  |  |  |  |  |  |  |  | x |  |
| Stand type |  |  |  |  |  |  |  |  |  |  |  |  |  |  |  |  |  |  |  |  |  |  |  |  | x |  |
| Stand age |  |  |  |  |  |  |  |  |  |  |  |  |  |  |  |  |  |  |  |  |  |  |  |  | x |  |
| Stand canopy density |  |  |  |  |  |  |  |  |  |  |  |  |  |  |  |  |  |  |  |  |  |  |  |  | x |  |
| Human Index |  |  |  |  |  |  |  |  |  |  |  |  |  |  |  |  |  |  |  |  | x |  |  |  |  |  |

The probability of a fire spreading and the damage it will cause are defined as fire risk [16,17]. Toward the end of the 20th century, remote-sensing infrared scanners began to be used to locate forest fires [40]. Today, thanks to the recurrence interval and wide-area imaging capabilities of satellite data, it has become possible to obtain important information about various fires. Technologies such as remote sensing and geographic information systems (GISs) have made significant contributions to fire prevention by providing capabilities for data collection, analysis, and mapping [16]. To model fire risk, it is essential to identify factors that significantly influence the probability of a fire [41]. In addition to the geographical characteristics, climate, etc., mentioned in the study, human factors are the main reasons affecting the outbreak and spread of fire in a region [42]. Moreover, many criteria are used in risk models, and they can be handled in groups, such as the distance to settlement structures and their type, as well as the amount and distribution of flammable materials, which were also mentioned in [17].

Fire risk maps for a region are created by combining factors that could lead to a fire [24]. For example, to create risk maps for forest fires, methods such as the analytic hierarchy process (AHP), fuzzy logic, goal programming (GP), artificial neural networks (ANN), and machine learning methods such as random forest and logistic regression are utilized while leveraging geographic information systems (GISs) [17].

The machine learning methods used in this study have been employed in recent years for such applications as fire forecasting, fire risk assessment, creating hazard risk susceptibility maps, and modeling fire behavior [43–45]. Classification through machine learning methods is becoming a commonly used approach in various fields. Many studies in the literature indicate that machine learning methods are efficient in classification problems, offering high accuracy and minimal deviation [46–51].

Lu et al. [46] developed a classification model aimed at predicting the fire risk in stadiums. The model used fire risk data from smart stadiums. The study concluded that the best performance was achieved with the gradient boosting model, with an F1 score of 81.9% and an accuracy of 93.2%

In their study on forest fire prediction, Pang et al. [47] collected data on fire hotspots, meteorological conditions, terrain, vegetation, and socioeconomics from various sources. Using these data, they developed models using an artificial neural network, radial basis function network, support vector machine, and random forest to identify the thirteen main causes of forest fires in China. The study reported that the prediction accuracies of the four forest fire prediction models ranged from 75.8% to 89.2%, and the area under the curve (AUC) values ranged between 0.840 and 0.960.

Kalantar et al. [52] conducted forest fire susceptibility prediction based on remote sensing data using resampling algorithms in machine learning models. The study concluded that the boosted regression tree model outperformed other models with the highest AUC value of 0.91. Additionally, they emphasized that the prediction performance of all models was improved when using the resampling process.

In Turkey, fires are on the rise due to the effects of global warming and uncontrolled urbanization. The year 2021 has gone down in history as a very bad year for Turkey in terms of fires. In the last 50-year period, major fires have been observed in 11 provinces. According to the records of the General Directorate of Forestry (OGM), 117,734 fire incidents have occurred from 1937 to the present day, resulting in the loss of a very large forest area of 1,851,476 hectares. This equates to an average loss of approximately 15.73 hectares per fire. In the last 10 years, Muğla (2716), Antalya (2446), İzmir (1649), and, relevant to this study, Istanbul (1493) have been the four provinces with the highest numbers of fires. These data are based on the OGM's 2022 records [53].

Istanbul has experienced numerous devastating disasters over time. The expectation of an earthquake is a topic frequently discussed in scientific circles today. Studies are being conducted at the urban scale concerning disaster preparedness and post-disaster recovery. One of the significant reasons for directing this study toward this area is the city's inherent risk. Fires are known to be a type of disaster that can occur in conjunction with

earthquakes [54]. However, the focus of this study is not solely on urban fires; it also aims to address the fire risk posed by the natural areas within and surrounding a city. When studies on fire risks in wildland–urban interface (WUI) areas first began, the extent of the threat that natural area fires posed to humans and settlements drew attention [11]. Therefore, the aim of this study is to visualize the risk dimension in WUI areas. As cities expand into "wildlands", they develop into WUIs, and the land there is converted into residential areas over time. In this respect, it is necessary to present the current risk distribution of WUIs in terms of fire in a region-specific manner, depending on factors with a high level of importance, to the attention of urban planners and city-related management boards. For this purpose, data collected through remote sensing techniques and tools were evaluated using GIS tools. The importance levels of the factors affecting the occurrence of fires were determined using machine learning methods, and a risk map was created using data obtained from both avenues.

## 2. Materials and Methods

### 2.1. Research Site and Data Collection

Istanbul is located at the junction of Europe and Asia. It consists of two peninsulas, Çatalca and Kocaeli. It is bordered by the Black Sea in the north and the Sea of Marmara in the south. It shares its borders with the provinces of Tekirdağ and Kocaeli. It is situated at a geographic location of 41.0122 latitude and 28.976 longitude (Figure 1). The city has a green vegetation cover consisting of forests, maquis, and various tree communities. Due to the climate, dry plant species are more prevalent in the northern regions. In terms of fire risk, consideration of this feature is important [55].

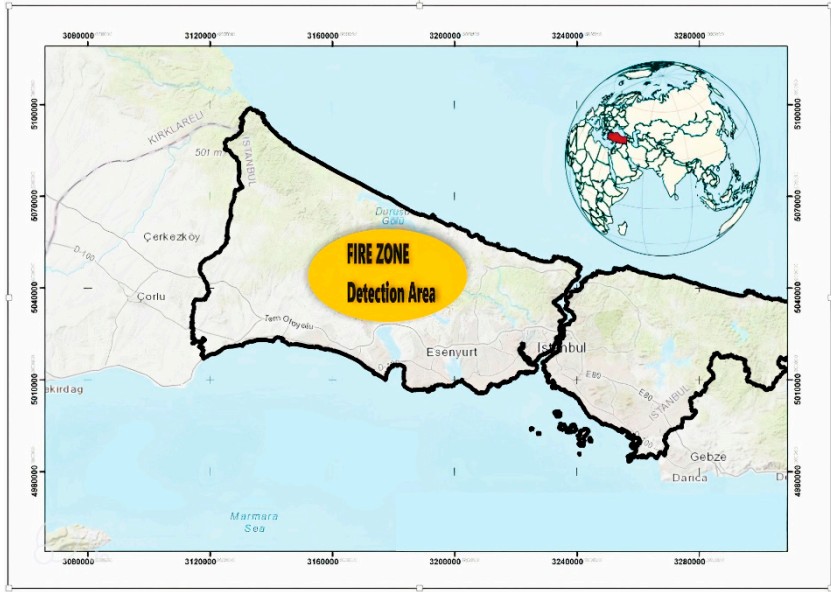

**Figure 1.** Map of the study area showing the location of Istanbul (produced using QGIS).

As an urban settlement, the region is located on the Marmara Sea coast in the south and on the Bosphorus coast in the east, with population densities in the range of 4000–50,000 people/km$^2$. The remaining areas appear to be intermixed between rural–urban areas, wildland–urban areas, agricultural space, peri-urban regions, and farms, sometimes forming surface boundaries. In addition to these, forests and similar wild areas can be seen extending toward the Black Sea to the north. Even in areas that appear to be entirely wild, there are various-sized points that are used by humans, referred to as anthropic space.

Geographic information systems significantly facilitate the evaluation and interpretation of spatial studies. Therefore, in this study, data obtained with remote sensing techniques were used along with geographic information systems (Figures 2 and 3).

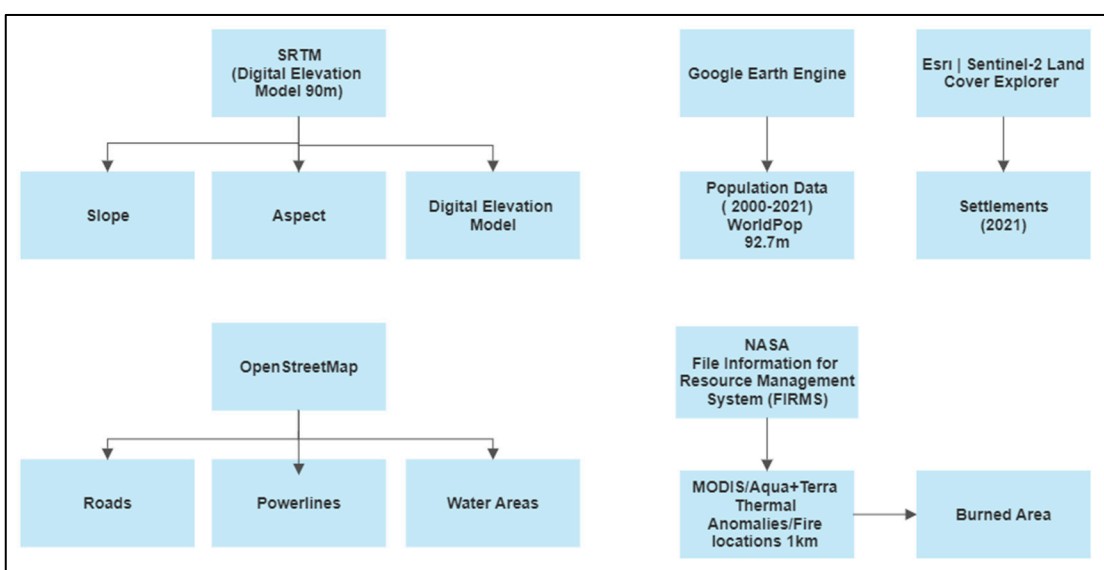

**Figure 2.** Collected data and their sources.

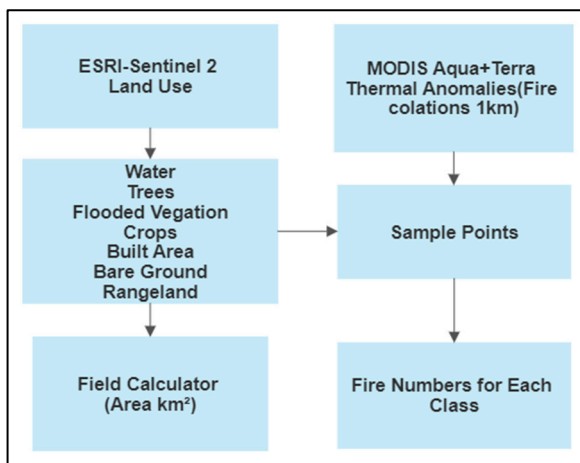

**Figure 3.** Land use/land cover data: sources and procedures.

In forest and rural fires, factors such as topography, climate, and vegetation type are considered crucial. However, for areas affected by human activities within the natural environment, factors such as distance to roads, population, building materials, and power lines or urban infrastructures are also taken into account [13]. Based on this information, the most preferred layers in the studies have been determined, in addition to the ones most necessary in terms of the characteristics of our study area. As a result, eight factors were identified: slope, aspect, elevation, population, distance to roads, distance to power lines, distance to settlements, and distance to water bodies. Only some of the data for these factors are given in Table 2, but the table shows data for the burnt areas of the region between 2000 and 2021 and nine parameters. These are also shown in Figure 2 along with their sources. The dataset in Table 2 was created for machine learning. Here, the "Count" column shows the number of samples created at 1 km intervals in the study area. A sample interval of 1 km was chosen, as this is compatible with the resolution of the FIRMS data. The rows of the factors contain the pixel values at the sample point. The "Fire status" was

obtained by assigning a value of 1 if a fire occurred at the sample point and 0 otherwise. The "Mean" column shows the average of all values determined for each factor. Some descriptive statistics are shown in the other columns.

**Table 2.** Statistical parameters of the dataset.

|  | Count | Mean | Std | Min | Max |
|---|---|---|---|---|---|
| Slope (SL) (°) | 3455 | 6.34 | 6.54 | 0.00 | 48.60 |
| Aspect (AS) (°) | 3455 | 134.16 | 109.03 | −1.00 | 356.55 |
| Digital elevation model (DEM) (m) | 3455 | 114.56 | 73.68 | 1.00 | 428.00 |
| Distance to power lines (DP) (m) | 3455 | 4175.25 | 3678.12 | 0.00 | 21801.60 |
| Population (PO) (people) | 3455 | 16.13 | 50.24 | 0.03 | 470.96 |
| Distance to roads (DR) (m) | 3455 | 145.61 | 183.30 | 0.00 | 2046.85 |
| Distance to water areas (DW) (m) | 3455 | 1939.42 | 1396.11 | 0.00 | 8547.64 |
| Distance to settlements (DS) (m) | 3455 | 543.93 | 799.76 | 0.00 | 5734.47 |
| Fire status (FS) | 3455 | 0.07 | 0.26 | 0.00 | 1.00 |

The layers created with the data of the factors considered in the study have different characteristics. Detailed feature information is given in Table 3 with the name of the factor data, the source, and the resolution information for a better understanding of the datasets. There are four main data sources, which are Open Street Map (OSM), the United States Geological Survey (USGS), Google Earth Engine (GEE), and the Living Atlas of the World (ArcGIS).

**Table 3.** Dataset features.

| Platform | Data | Source | Resolution |
|---|---|---|---|
| OSM | Road | http://overpass-turbo.eu | |
|  | Water Areas | http://overpass-turbo.eu | |
|  | Power Line | http://overpass-turbo.eu | |
| USGS | SRTM | http://earthexplorer.usgs.gov/ | 90 m |
| ArcGIS | Land Cover | https://livingatlas.arcgis.com/ landcoverexplorer | 10 m—2021 |
| GEE | WorldPop | ee.ImageCollection ("WorldPop/ GP/100m/pop") | 92.7 m |
| FIRMS | MODIS Aqua+Terra Thermal Anomalies (Fire Locations) | https://firms.modaps.eodis.nasa.gov/ | 1 km |

*2.2. Method*

2.2.1. GIS-Based Processes

First, the land use/land cover (LULC) distribution of the region was examined. The LULC data had a resolution of 10 meters and included information for 7 land classes (water, trees, flooded vegetation, crops, built area, bare ground, rangeland). These data were produced by Impact Observatory, Esri, and Microsoft from the ESA Sentinel-2 data source [56]. The areas of the above-mentioned land classes were calculated. Initially, the data were downloaded from open-access sources and then color-coded in GIS software. Subsequently, the number of fires occurring in these calculated areas was identified. These operations can be easily performed using GIS tools, such as Field Calculator and Sample Point. The calculated area and the numbers of fires that occurred in these areas are presented in Table 4. The produced LULC map is shown in Figure 4.

**Table 4.** Land-use classes and fires occurring in these areas.

| Classification | Number of Fires | Area (km$^2$) | Area (%) |
|---|---|---|---|
| Water | 0 | 112,845 | 3.18 |
| Trees | 11 | 1354,764 | 38.19 |
| Flooded Vegetation | 0 | 1782 | 0.05 |
| Crops | 168 | 954,511 | 26.91 |
| Built Area | 302 | 891,761 | 25.14 |
| Bare Ground | 0 | 21,838 | 0.62 |
| Rangeland | 28 | 209,752 | 5.91 |

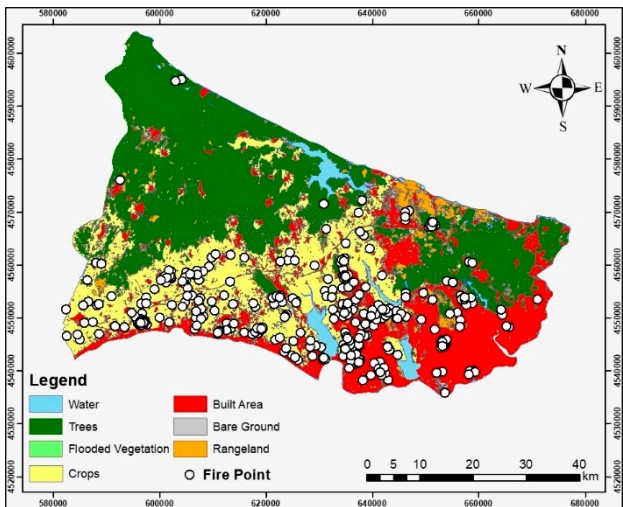

**Figure 4.** LULC map and fire points of Istanbul.

The GIS operation steps of the study are shown in Figure 5. Open-source software and platforms were preferred in each GIS operation step.

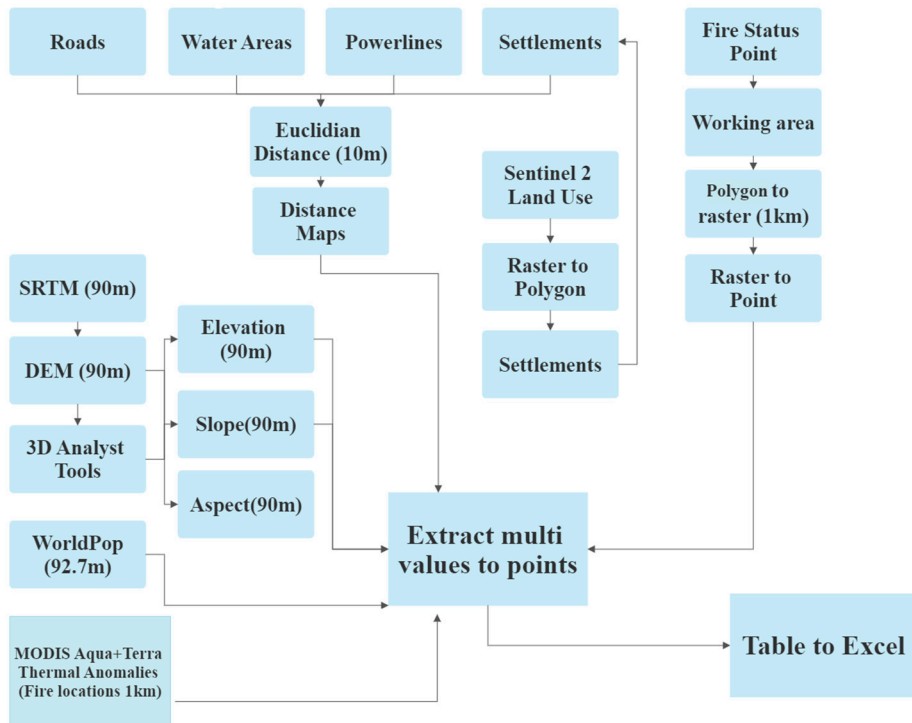

**Figure 5.** Flowchart of GIS operation.

A map of the European side of Istanbul map was created in the software as vector data with a resolution of 1 km. It was transformed from vector data into a raster data structure using the "polygon to raster" tool from the GIS tools. The "raster to point" tool was used to create point data at 1 km intervals.

From another data source, Open Street Map (OSM), data for platform products, distance to roads, water areas, and lower lines were obtained via an open-access web platform [57] using the code blocks given in Appendix A. The obtained findings are shown in Figure 6. The images created were in the Keyhole Markup Language (KML) file format and were created as a layer in the GIS software using the "KML to Layer" operation. In this study, it was not possible to compare the LULC map, which is shown in Figure 22, and the risk map, which is shown in Figure 4, with a numerical model. It would be very useful to propose a model for this in the next phase.

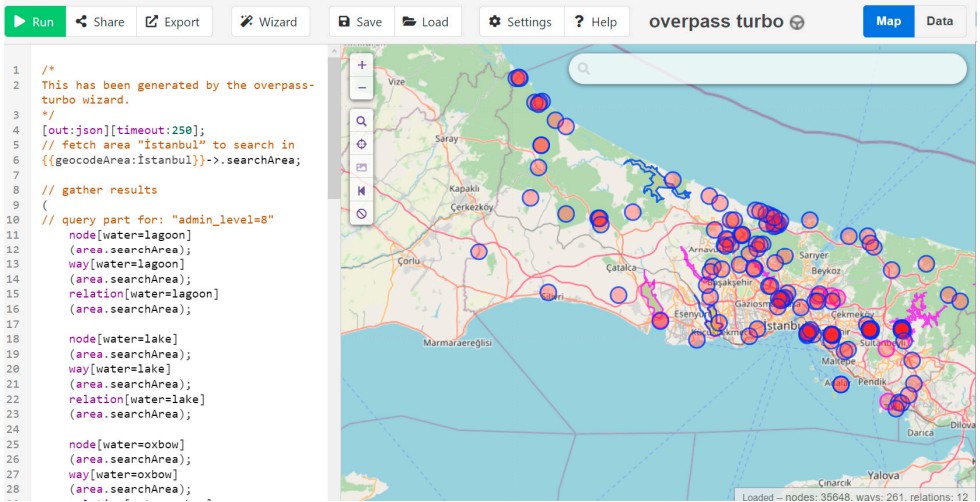

**Figure 6.** OSM mapping platform.

The "3D Analyst Tools" tool in the GIS software was used to obtain the slope and aspect data of the study area. The elevation data were used to calculate the slope value. With the help of the DEM data, an aspect map representing the interaction direction of the terrain surfaces with the sun was obtained. Figure 7 shows the slope map, and Figure 8 shows the aspect map.

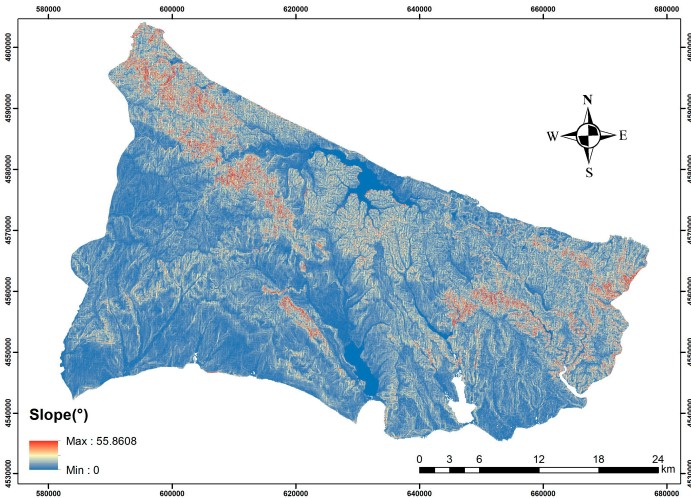

**Figure 7.** Slope map of the study area.

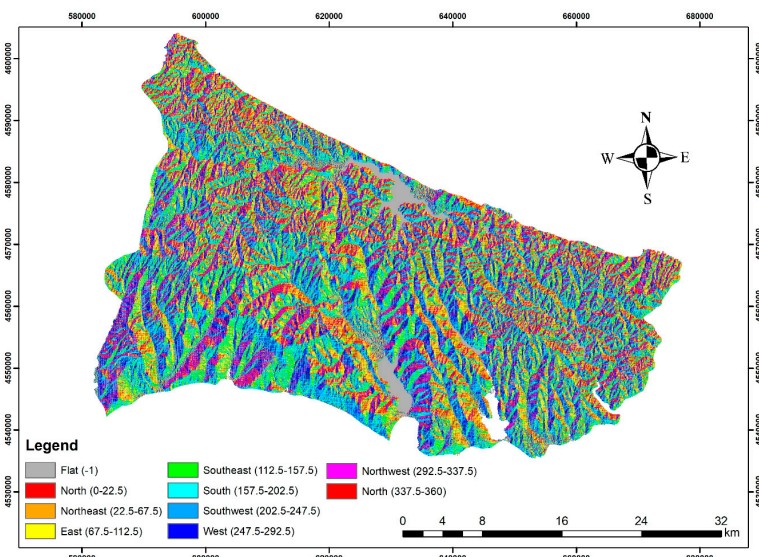

**Figure 8.** Aspect map of the study area.

The slope, as seen in Table 1, is a factor used as a criterion in all examined studies. In the majority of studies, particularly where forest fires are concerned, an increase in slope means increased risk, as the flames meet the fuel more quickly and spread rapidly as the slope steepens due to wind speed [14,17,23,24,29,38]. However, the slope is not a risk factor that causes a fire to start but is one that intensifies or accelerates a fire once it starts. In some studies, the slope is considered in relation to agricultural activities [27]. Since agricultural activities are generally carried out on low-slope lands and are not preferred as the land becomes steeper, human factors causing fire will decrease as the slope increases. From this point of view, an increase in slope reduces the risk. This perspective is more appropriate for the region covered in this study. Figure 7 shows that there are very few areas where the slope is as high as 55.86° and that the region mainly consists of low-slope lands ranging from 0 to 8.32°. In the LULC map, it can also be easily seen that agricultural activities are carried out in these low-slope areas (Figure 4). For these very apparent reasons, agricultural activities were considered as fire-starting factors, and this study adopted the approach that the risk increases as the slope decreases.

Aspect is the information about which direction the land parcels face. In the Northern Hemisphere, slopes facing the south are exposed to sunlight and radiation heat for longer and with greater strength compared to those facing the north [14,16,23,28,38]. Regarding aspect, south-facing slopes are considered to have the highest risk, while north-facing slopes are considered risk-free because they are not directly exposed to sunlight and are exposed to moist northern winds coming from the sea. Intermediate values are given for other directions.

Another data source in this study was the Shuttle Radar Topography Mission (SRTM), which provided topographic data. A map was created in the GIS software environment using DEM (digital elevation model) data obtained from this source. This map is presented in Figure 9.

As the elevation increases, the likelihood of precipitation increases due to increased humidity and decreased temperature, resulting in lower fire risk than that at lower elevations [14,17,22,29,39]. Therefore, lower elevations were considered to have a higher risk, while higher elevations were considered to have a lower risk.

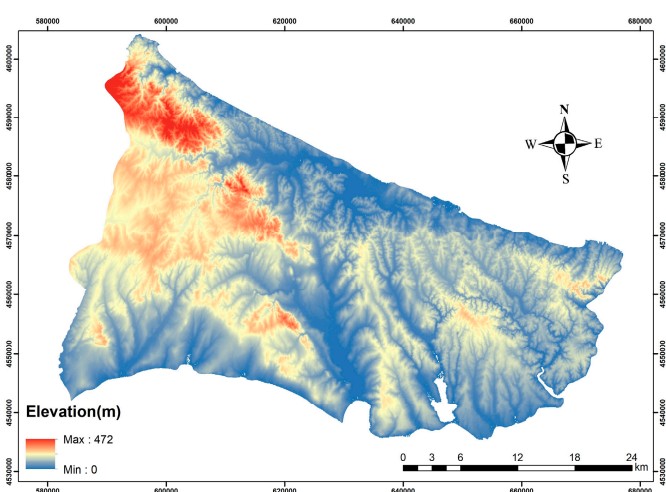

**Figure 9.** Digital elevation model (DEM).

For the determination of the residential areas, land-use data obtained with Sentinel-2 by Esri were preferred, and the residential areas covering the year 2021 were identified [58]. The data obtained were used to create the map in Figure 10 by using the Euclidean distance method, which measures the distance between two points to generate distance maps. The process was converted into vector data format using the "raster to polygon" tool in Esri Sentinel-2, and residential areas were selected. Subsequently, distance maps at 10 m intervals were obtained using the "Euclidean distance" tool.

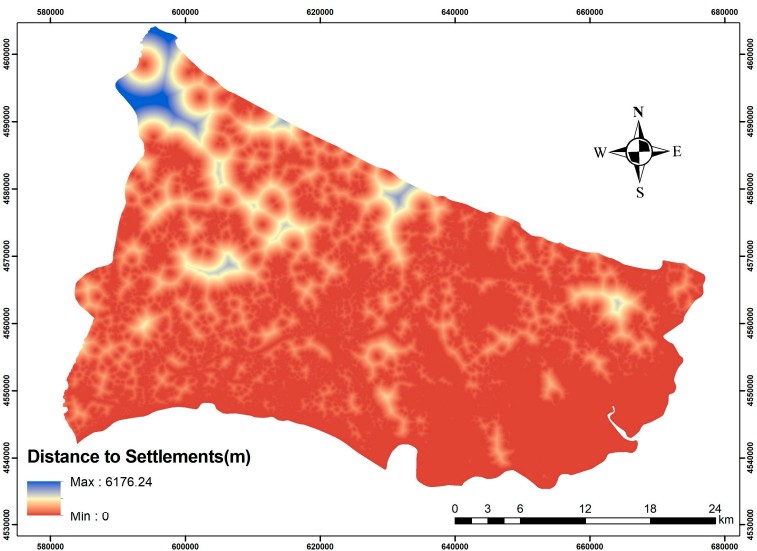

**Figure 10.** Distance to settlements.

Natural areas have fire risk due to factors such as population density, accidents, negligence, various human activities, slash-and-burn agriculture, and clearing land for settlement, among others [15–17,19,26,29,39]. The closer to residential areas, the higher the risk; the risk decreases as the distance increases.

For population density, WorldPop data covering the years 2000–2021, which are used globally to map population distribution, and the open source platform GEE were used to obtain information. These data were used to create the map in Figure 11.

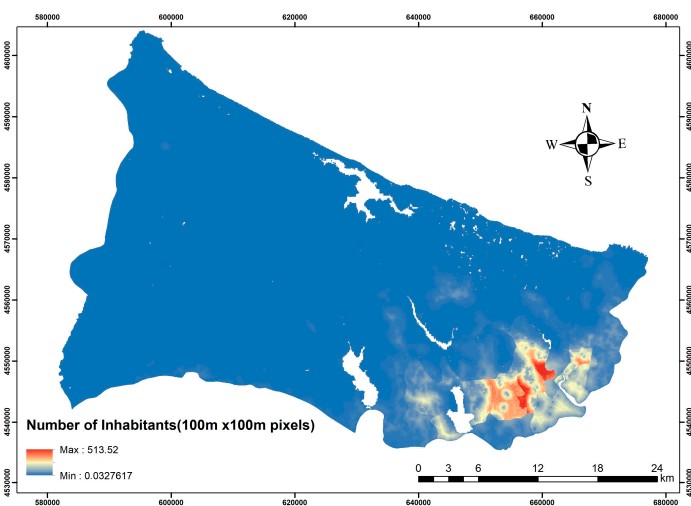

**Figure 11.** Population density map of the study area.

The distance maps for road proximity, water areas, and power lines were obtained using the "Euclidean distance" function after extracting the corresponding layers from the OSM dataset. The resulting distance maps are shown in Figures 12–14.

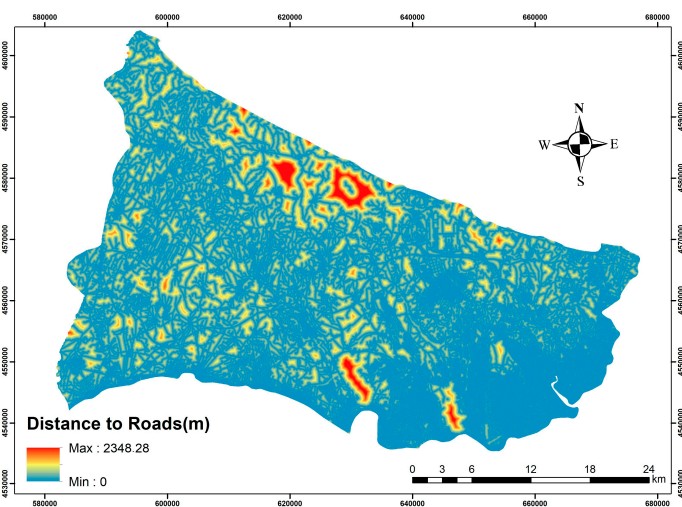

**Figure 12.** Distance to roads.

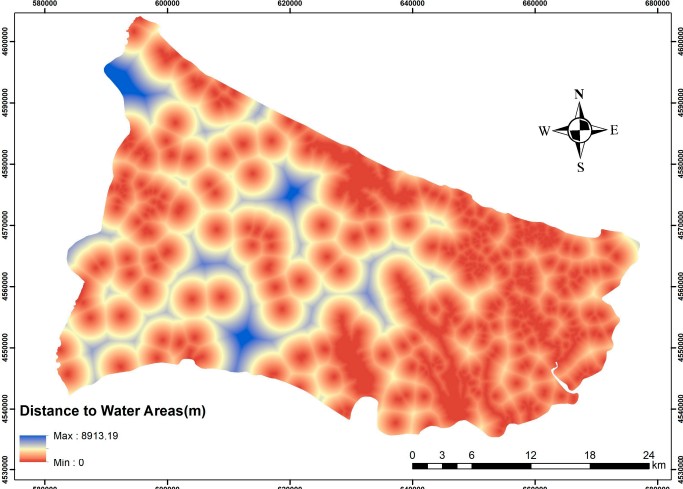

**Figure 13.** Distance to water areas.

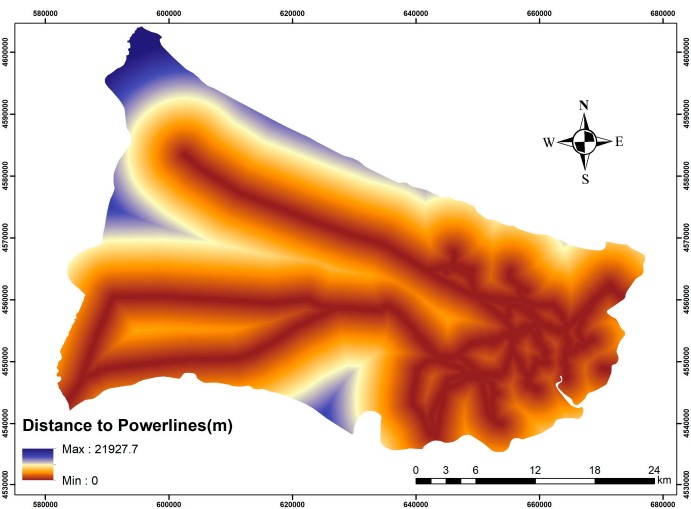

**Figure 14.** Distance to power lines.

In short, the likelihood of fire increases wherever there are roads and energy lines. This is because roads increase human mobility toward an area, carrying all kinds of risks for various parts of nature [15–21,28,29,38,39]. Power lines are a very important factor for fire risk because, depending on meteorological conditions, they can start a fire, and when these two meet, the spread of fire can be facilitated. Examples of meteorological conditions are the expansion and sagging of the conductors in the lines in hot and dry weather conditions and the collision of the conductors with each other with the effect of the wind. In this way, when a fire starts, it can spread rapidly due to the effect of dry air and wind. For this reason, this tends to create a larger burnt area than fires started by other sources [59]. Power transmission lines play an important role in forest and WUI fires as a human-made factor because faults in the lines can, at the same time, also cause fires, including in cities, regardless of meteorological conditions [60,61]. Proximity to water, on the other hand, is where the risk is lowest, and even in the event of a fire, it increases the strength and speed of intervention [19,20,38].

Due to unit and value differences between layers, the values of each layer were standardized through normalization to between 0 and 1 (n).

$$n_i = \frac{raster - raster(min)}{raster(max) - raster(min)} \tag{1}$$

Here, $n_i$ represents the i-th layer value in each layer. The values correspond to increasing risk as the values approach from 0 to 1, but the opposite was applied for slope, height, distance to water, and distance to road and power lines, as explained for each of the reasons in this study.

In this study, two types of processes were performed. The first type was point-based processes. Using point data at 1 km intervals, the "extract multi values to points" tool was used to obtain the digital number (DN) values of pixels, and from these, a matrix was created. An Excel table was created using the "Table to Excel" tool. The DN value matrix obtained for each layer was used as a data source in the next process step, which involved machine learning methods, and variable importance (VI) values related to fire were obtained. Point-based processes facilitated risk assessments of the layers. The second type was pixel-based processes. Pixel-based processes also comprise spatial risk assessments. All layers related to fire (DEM, slope, aspect, etc.) were in raster format. The "MODIS/Aqua+Terra Thermal Anomalies/Fire locations 1km" dataset from NASA's File Information for Resource Management System (FIRMS) platform was used for the detection of burned areas covering the years 2000–2021. These data were obtained from Terra and Aqua satellite images. The map created for the burned area is presented in Figure 15.

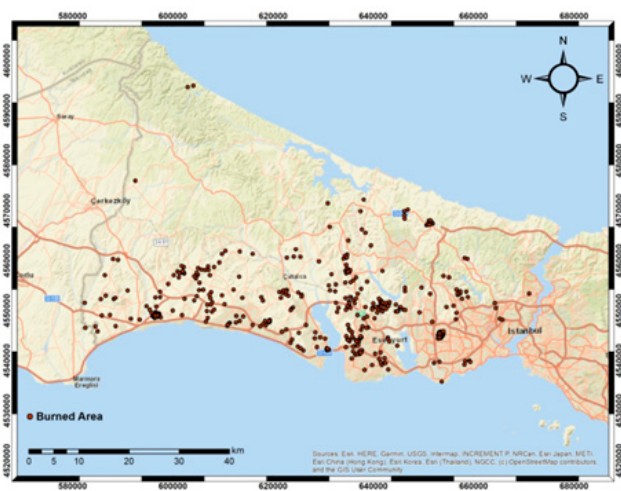

**Figure 15.** Fire locations (between 2000 and 2021).

2.2.2. Machine Learning Methods

In cases where there is an imbalance between classes in a dataset, machine learning algorithms usually make biased decisions in favor of the majority class. Since there was an imbalance between classes in the dataset used in this study, the frequency of the majority class was balanced with the random sampling method to match the frequency of the minority class. Thus, the aim was to make the evaluation metrics more consistent.

The dataset created within the scope of this study was randomly divided into a training set (70%) and a test set (30%). Numerous classification models were established on the training set for fire risk prediction. These included the extra trees, random forest, light gradient boosting, gradient boosting, extreme gradient boosting, decision tree, AdaBoost, and k-neighbors machine learning classification methods. The best results were obtained using the random forest, extreme gradient boosting, and light gradient boosting methods. A schematic view of the classification based on machine learning is given in Figure 16.

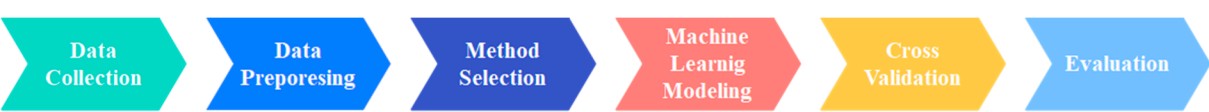

**Figure 16.** Schematic view of machine-learning-based modeling.

2.2.3. Random Forest (RF)

The random forest method, a community-based classifier developed by Breiman [62], is frequently used in classification and regression problems. The aim of the random forest method, an integrated algorithm of the bagging method, is to combine the decisions of a series of classifiers through weighted or unweighted voting [51].

2.2.4. Extreme Gradient Boosting (XGB)

The XGB method, a machine learning method based on gradient boosting developed by Chen and Guestrin [63], uses gradient descent in a decision tree to create an optimal model. As a method based on ensemble learning, XGB sequentially builds multiple decision trees considering the impact of high-performance decision tree models, aiming to minimize the errors made by previous decision trees through subsequent decision trees [51].

2.2.5. Light Gradient Boosting (LGB)

Light gradient boosting is an improved version of the gradient learning framework based on decision trees and the idea of "weak" learners. Following its development by Microsoft in 2017, LGB has been widely applied in many fields as a result of its high

prediction accuracy, fast computing speed, and excellent ability to minimize overfitting problems [64].

### 2.2.6. K-Fold cross-Validation

In the k-fold cross-validation method used in this study, the data were randomly grouped and divided into "k" subgroups. One of these was used for testing, and the remaining "k-1" were used for training. This process was repeated "k" times. The average of the results determined the accuracy of the method [50]. The diagram of the k-fold cross-validation is shown in Figure 17.

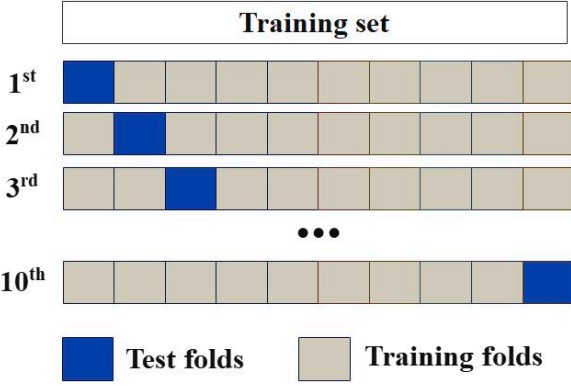

**Figure 17.** K-fold cross-validation diagram.

### 2.2.7. Model Evaluation

In this study, we evaluated the performance of the trained models based on the accuracy, precision, recall, F1 score, AUC score, and receiver operating characteristic (ROC) curve, which were calculated according to a widely used confusion matrix and are expressed in Table 5 using Equations (1)–(4) [46]. The ROC curve is the curve of the true positive rate and false positive rate at different classification thresholds. It starts at (0,0) and ends at (1,1). A good model produces a curve that rapidly goes from 0 to 1. The AUC (area under the ROC curve) summarizes the ROC curve as a single number. The AUC value varies between 0.5 and 1. The highest AUC value indicates an excellent measure of separability, while the lowest AUC value indicates the worst measure of separability.

$$Accurarcy = \frac{TP + TN}{TP + TN + FP + FN} \tag{2}$$

$$Precision = \frac{TP}{TP + FP} \tag{3}$$

$$Recall = \frac{TP}{TP + FN} \tag{4}$$

$$F1 = 2\frac{Precision \ x \ Recall}{Precision + Recall} \tag{5}$$

**Table 5.** Classification confusion matrix.

| | | Predicted Value | |
|---|---|---|---|
| | | **Fire (Class 1)** | **Non-Fire (Class 0)** |
| **Actual value** | Fire (class 1) | True Positive (TP) | True Negative (TN) |
| | Non-Fire (class 0) | False Positive (FP) | False Negative (FN) |

The data collection methods and analytics described up to this point were used to generate a risk map. For this, the values associated with each layer created by individual factors were combined with the feature importance (FI) values obtained through machine learning methods. These values served as coefficients (a, b, c, ..., h) and were multiplied in a raster calculation to yield risk scores. The mapping was organized according to five levels of risk.

$$Risk\ Score = SL \times a + AS \times b + DEM \times c + DP \times d + PO \times e + DR \times f + DW \times g + DS \times h \quad (6)$$

## 3. Results and Discussion

In the literature, it has been highlighted that various machine learning algorithms are generally quite successful due to their ability to learn from and model data, and they often yield better results than those of traditional statistical approaches [65]. In this study, the results obtained from the three most successful models in terms of feature importance—random forest, extreme gradient boosting, and light gradient boosting—were compared with the available data. The quality parameters used in model validation are provided in Table 6. Upon examining the most important validation parameters, such as accuracy and AUC (area under the curve), the best results were obtained from the random forest model. The lowest results were recorded for the LGB model with respect to the accuracy parameter and for the XGB model with respect to the AUC parameter. However, the differences between them were negligible. Nevertheless, the RF model, which possessed a higher recall value and F1 score, distinctly separated itself in terms of accuracy from the closely matched XGB and LGB models. It should be noted that the other two models also showed remarkable performance, but at this stage, the best RF model was selected, and the risk assessment was calculated according to the results of the RF model.

**Table 6.** Evaluation parameters for the classification models.

| Model | Accuracy | AUC | Recall | Precision | F1 |
|---|---|---|---|---|---|
| Random Forest (RF) | 0.6975 | 0.7606 | 0.7559 | 0.6809 | 0.7127 |
| Extreme Gradient Boosting (XGB) | 0.6972 | 0.7607 | 0.6974 | 0.7006 | 0.6956 |
| Light Gradient Boosting (LGB) | 0.6715 | 0.7429 | 0.6745 | 0.6778 | 0.6725 |

Rodriguez and Riva (2014) used machine learning models to assess human-induced wildfires in Spain between 1988 and 2007 and to predict fire risk. The whole country, when considered, does not entirely consist of natural areas and forests but includes regions with different characteristics, such as wildland–urban interface (WUI), rural–urban interface (RUI), and wildland-agricultural interface (WAI) regions. In an earlier study, factors such as population density, energy lines, railways, and agricultural vehicle density were considered [66]. Except for the railway and agricultural vehicle density, the factors and the manner of field examination were quite similar to those of our study. In the earlier study, it was revealed that the RF model was the best model with an AUC value of 0.746. In the present study, the RF model also provided the best result with an AUC value of 0.760. The 10-fold cross-validation method was used to validate the model. The average value of the obtained results was taken into account. The k-fold validation results for all three models are given in Table 7. Upon reviewing the table, it is observed that the standard deviation (std) values for accuracy and AUC were low. This was true for all three models.

The fire risk prediction capacity of the classification models was tested using ROC analysis. The AUCs of the ROC graphs for the RF, XGB, and LGB models were 0.73, 0.74, and 0.76, respectively, for the test data (Figures 18–20).

**Table 7.** K-fold validation accuracy and AUC predictions for the classification models.

| | RF | | XGB | | LGB | |
|---|---|---|---|---|---|---|
| **Fold** | **Accuracy** | **AUC** | **Accuracy** | **AUC** | **Accuracy** | **AUC** |
| 0 | 0.9298 | 0.6373 | 0.9050 | 0.7278 | 0.9174 | 0.7244 |
| 1 | 0.9339 | 0.7425 | 0.9174 | 0.7746 | 0.9132 | 0.7569 |
| 2 | 0.9298 | 0.7586 | 0.9132 | 0.7762 | 0.8967 | 0.7710 |
| 3 | 0.9421 | 0.8424 | 0.9091 | 0.8434 | 0.9050 | 0.8165 |
| 4 | 0.9256 | 0.8203 | 0.9215 | 0.7619 | 0.9091 | 0.7793 |
| 5 | 0.9215 | 0.7975 | 0.9050 | 0.7371 | 0.9050 | 0.7470 |
| 6 | 0.9256 | 0.8259 | 0.9132 | 0.7956 | 0.9256 | 0.8058 |
| 7 | 0.9256 | 0.7240 | 0.9174 | 0.7612 | 0.9132 | 0.7269 |
| 8 | 0.9253 | 0.6611 | 0.9170 | 0.6799 | 0.9087 | 0.6893 |
| 9 | 0.9253 | 0.7064 | 0.9087 | 0.6197 | 0.9170 | 0.6657 |
| Mean | 0.9285 | 0.7516 | 0.9127 | 0.7478 | 0.9111 | 0.7483 |
| Std | 0.0056 | 0.0669 | 0.0054 | 0.0590 | 0.0077 | 0.0456 |

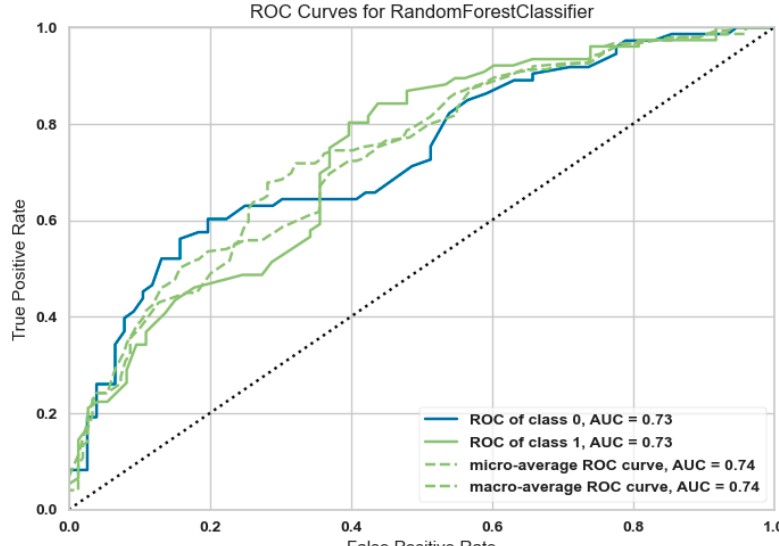

**Figure 18.** ROC curves for the RF classifier.

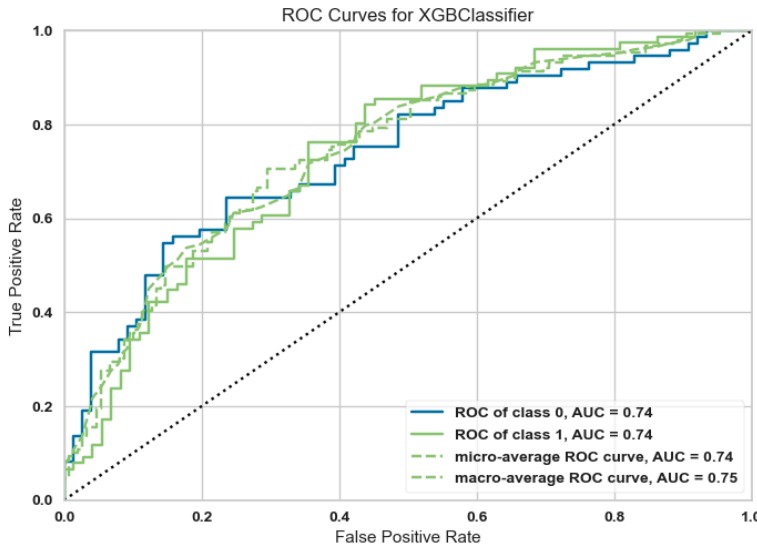

**Figure 19.** ROC curves for the XGB classifier.

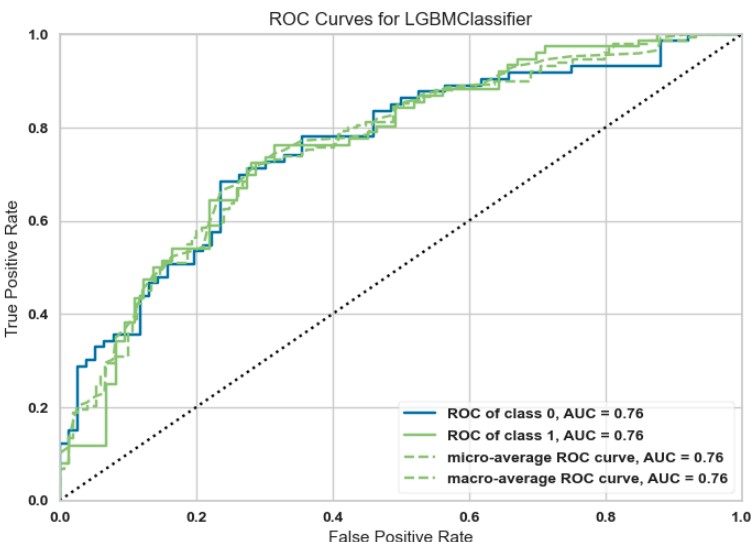

**Figure 20.** ROC curves for the LGB classifier.

The RF (Figure 21) and XGB models indicated that the population factor had the highest importance. The LGB model, on the other hand, emphasized the distance to water as the most important, ranking the population factor in fourth place. Proximity to power lines was the second most important factor in both the RF and XGB models, while distance to residential areas ranked third. The LGB model also placed proximity to power lines as the second most important factor, and it raised elevation, a topographic factor, to third place.

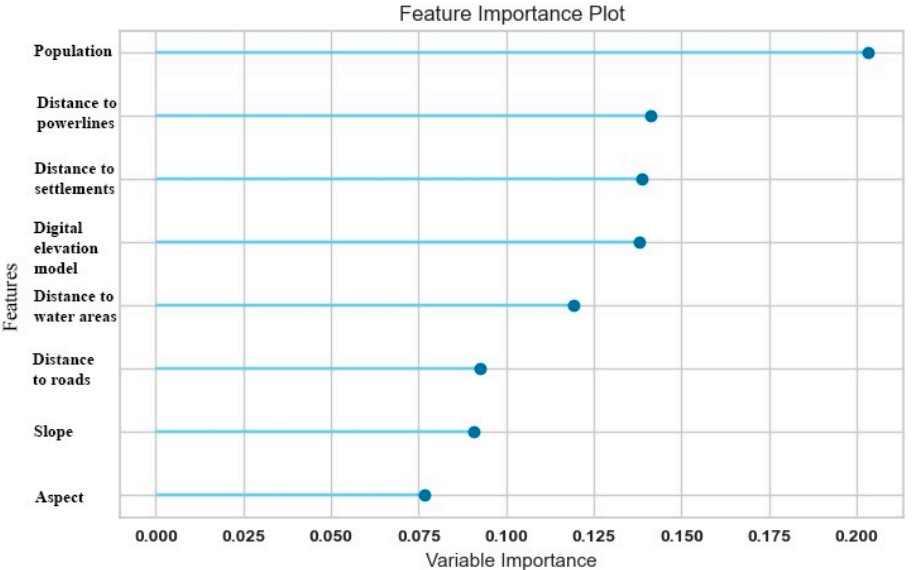

**Figure 21.** Feature importance for the RF classifier.

In the classification of the RF and XGB models, the factor of highest importance was the population by a significant margin, followed by proximity to power lines and distance to residential areas as the second most important factors; these are generally considered initiating factors in fires. Conversely, in the LGB model, the distance to water, which was the most important factor, and elevation, which was the third most important, are not typically initiating factors in fires. Proximity to power lines, the second most important factor in the LGB model, is a potential initiating factor in fires. Moreover, elevation has a direct impact on temperature, humidity, and wind. As elevation increases, the likelihood of precipitation generally rises, reducing the intensity of fires [67]. The LGB model ranked a

factor with a negative correlation to fire probability as third in importance. This factor was fourth in importance in the RF model and fifth in the XGB model, and they did not have the same value as in the LGB classification chart. Overall, the RF and XGB models appeared to have a closer alignment in their rankings of the importance of factors. However, we see that all three models yielded the same result for proximity to power lines. Power lines pass through natural areas of various properties that many city infrastructures do not even reach, meet with other secondary factors, and become some of the well-known causes of fires in natural areas [68]. The results of these models, which are very successful in classification and importance ranking, may vary according to the size and nature of the dataset and the type of problem. In this study, these models were required to determine the weights of the factors involved in the calculation of the total risk score and included as variables in Formula 6. Generally, in such cases, it is much more accurate to obtain the best result by making some comparisons and to make decisions with parameters—as shown in Tables 6 and 7, which reveal the performance of the models—instead of through subjective decisions. The comparisons show that the performances of the three models were close to each other, but RF was relatively the best. The feature importance values and the ranking revealed by the RF model were the same as those of the XGB model, and these models supported each other. The RF (random forest) model is generally able to achieve high accuracy, and it can do so with fewer feature variables. Fewer parameters actually mean easier calibration [65,67]. Nevertheless, no single model is perfect enough to always make accurate predictions and should, therefore, be evaluated in conjunction with other models.

The relationship between LULC (land use/land cover) and fires that occurred in Portugal, France, Italy, and Greece in the last 20 years was examined. This revealed a significant loss in forest and agricultural areas, an increase in shrublands, and that most fires occurred in coniferous forests. Strong correlations between LULC and fires have previously been reported [69]. LULC has been considered a factor in some wildfire risk assessment studies [14,28,30,34,40,42]. Risk assessments have been conducted using the analytical hierarchy process (AHP) method and, in one case, TOPSIS [28]. However, the inclusion of LULC data in other datasets often leads to deviations due to resolution limitations in GIS processing. Therefore, LULC was considered separately. Figure 4 clearly shows that the distribution of fires between 2000 and 2021 on the LULC map indicated high fire sensitivity in WUI (wildland–urban interface) areas. Table 4 reveals that almost half of the fires occurred in agricultural crops, which comprised 27% of the region, as opposed to building zones, which covered 25%. These were not interface areas where wildland and urban settlements were intertwined. In forested areas, comprising 38% of the region, the number of fires was quite low. The RF and XGB models, which were the most successful in classifying factors affecting fire risk, corroborated the low incidence of fires in this region by not giving high importance to topographic features.

The coefficients in Formula 6, from which Formula 7 was derived, were determined through machine learning methods. The risk map created from Formula 7 is presented in Figure 22, along with the distribution of fire points.

$$Risk\ Score = SL \times 0.093630 + AS \times 0.086122 + DEM \times 0.121648 + DP \times 0.132680 + PO \times 0.221696 \\ + DR \times 0.085067 + DW \times 0.130440 + DS \times 0.128717 \qquad (7)$$

If one recalls the areas covered by cultivated crops on the land use/land cover (LULC) map (Figure 4), it is noticeable that these areas were distributed across medium- and high-risk zones on the risk map, in which a significant number of fires had also occurred. The distribution of risk levels across areas and the number of fires can be seen in Table 8. It is understood that the number of fires shows a concerning increase as the risk level increases. Excluding the medium-risk level, if we combine the others to reduce the levels to three, approximately 1300 km$^2$ of the area is at low risk and, according to the table, 40 fires have occurred. Again, a nearly similar area (~1255 km$^2$) is at medium risk, with 184 fires having occurred. An area of about 883 km$^2$ was identified as carrying a high level of risk with a total of 285 fires.

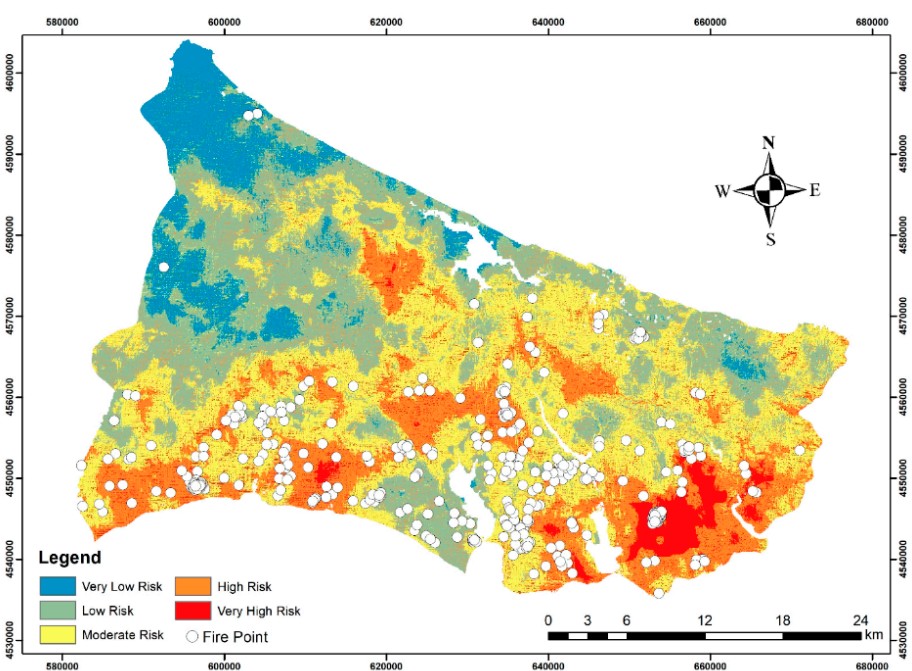

**Figure 22.** Risk map and fire points.

**Table 8.** Total area and fire counts by risk level.

| Risk Level | Total Number of Pixels | Area (km$^2$) | Number of Fires |
| --- | --- | --- | --- |
| Very Low Risk | 558,103 | 368,031 | 3 |
| Low Risk | 1,421,297 | 937,248 | 37 |
| Moderate Risk | 1,902,320 | 1254,450 | 184 |
| High Risk | 1,007,792 | 664,570 | 139 |
| Very High Risk | 331,371 | 218,517 | 146 |

Human population and structures that develop with the population, such as energy lines, roads, and settlements, have been determined to be factors of high importance for rural area and forest fires, and it is clear that these factors are also important parameters in planning cities. Based on this, the fire risk in planning settlements that tend to expand toward natural areas can be evaluated with this study model made for the European side of Istanbul and can be safely used in decision-making processes for urban planning. The proposition that this study puts forward for designs in urban planning works is to pull back the approach of determining construction and infrastructure opportunities aimed at reaching sufficient capacities to meet a population size as much as possible. This shows that the dominant influence of the population and its related factors (DS, DP, DR) in planning studies should be reduced and should be included as parameters to be kept under control.

A model for the probability of fire occurrence specific to the European side of Istanbul is presented as a result of this research. This region is characterized by its diversity of land use, including rural–urban and natural–urban amalgamations and areas used for agricultural purposes, farms, and cultivated fields. Therefore, predicting fire risk is essential. As a northward shift of the city is being considered to distance it from the fault line passing through the Marmara Sea due to expected earthquakes, incorporating fire risk into new settlement plans is crucial for developing sustainable residential areas.

## 4. Conclusions

The factors that play a role in fires that occurred in a certain time period were classified in terms of importance through machine learning algorithms. Among the RF, XGB, and LGB models, which have different bases used in these classifications, it was observed that the community-based RF model performed with high accuracy and prediction power, and

the XGB model based on decision trees gave results close to those of the RF model. Thus, reliable results were obtained with the determinations of different models that supported each other in identifying effective factors. This study was carried out using resources accessible to everyone, with data collection from completely open source platforms instead of data that generally take a long time to access and collect with software, languages, and programs. Therefore, every aspect can be examined, and it is possible to access the details of the data, software, and method. While this gave results compatible with those of some similar studies conducted earlier, where the same models were used, higher values were achieved for parameters such as the AUC, F1 score, and accuracy.

The study area was also examined in terms of land use/land cover (LULC), revealing that urban settlements tend to expand into natural areas in a manner that is more "intermixed" rather than "interfacial" in nature. This was evident from the amount of area covered by agricultural activities, cultivated lands, and, to some extent, pastures. The LULC map further indicated that these areas have been significantly impacted by fires. Utilizing machine learning for high-accuracy classification, it was determined that anthropogenic factors hold high significance. Compared to topographic factors, these are more risk-prone in terms of fire initiation. Topographic factors, on the other hand, influence the spread of a fire after its onset, either exacerbating or mitigating the risk. In the context of the study area, factors affecting fire initiation and those affecting its spread were distinctly separated based on their level of importance. Since the resulting risk map distributed the risk degrees over the region, it was possible to use it by overlapping it with settlement maps of the same regions. Thus, highly accurate parameters were identified, providing valuable insights for future urban planning and development, especially when considering a change in approach. In this study, it was not possible to compare the LULC map, which is shown in Figure 4, and the risk map, which is shown in Figure 22, with a numerical model. It would be very useful to propose a model for this in the next phase.

**Author Contributions:** Conceptualization, E.A. and K.A; software, E.A. and A.K.; validation, A.K.; investigation, K.A., İ.Y., and E.A.; resources, E.A. and İ.Y.; data curation, E.A. and A.K.; formal analysis, E.A. and A.K.; writing—original draft preparation, K.A. and A.N.A.; writing—review and editing, K.A., A.K., and A.N.A. All authors have read and agreed to the published version of the manuscript.

**Funding:** This research received no external funding.

**Institutional Review Board Statement:** Not applicable.

**Informed Consent Statement:** Not applicable.

**Data Availability Statement:** Not applicable.

**Conflicts of Interest:** The authors declare no conflict of interest.

## Appendix A. OSM Code Blocks

| Road | Water Areas | Power Line |
|---|---|---|

```
*/
[out:json][timeout:250];
// fetch area "İstanbul" to search in
{{geocodeArea:İstanbul}}->.searchArea;
// gather results
(
// query part for: "admin_level=8"
node[highway=motorway]
(area.searchArea);
way[highway=motorway]
(area.searchArea);
relation[highway=motorway]
(area.searchArea);

);
// print results
out body;
>;
out skel qt;
```

*\* The layers below, such as "lagoon" and "lake", have been obtained with the same method used in the code block above.*
*node[highway=trunk]*
*node[highway=primary]*
*node[highway=secondary]*
*node[highway=tertiary]*
*node[highway=unclassified]*
*node[highway=residential]*

```
*/
[out:json][timeout:250];
// fetch area "İstanbul" to search in
{{geocodeArea:İstanbul}}->.searchArea;

// gather results
(
// query part for: "admin_level=8"
node[water=lagoon]
(area.searchArea);
way[water=lagoon]
(area.searchArea);
relation[water=lagoon]
(area.searchArea);

node[water=lake] node[water=oxbow]

);
// print results
out body;
>;
out skel qt;
```

*\* The layers below, such as "lagoon" and "lake", have been obtained with the same method used in the code block above.*
*node[water=oxbow]*
*node[water=rapids]*
*node[water=river]*
*node[water=stream]*
*node[water=river]*
*node[water=stream_pool]*
*node[water=reservoir]*
*node[water=drain]*

```
*/
[out:json][timeout:250];
// fetch area "İstanbul" to search in
{{geocodeArea:İstanbul}}->.searchArea;
// gather results
(
// query part for: "admin_level=8"
node[power=line]
(area.searchArea);
way[power=line]
(area.searchArea);
relation[power=line]
(area.searchArea);
);
// print results
out body;
>;
out skel qt;
```

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
