# Peer review of "Assessing Fire Risk in Wildland–Urban Interface Regions Using a Machine Learning Method and GIS data: The Example of Istanbul’s European Side"

_fire, doi:10.3390/fire6100408_

Round 1

Reviewer 1 Report (Previous Reviewer 1)

Despite the improvements made to the manuscript, there are still many significant changes that need to be made.

- The introduction is still repetitive, e.g. 82-92 and 107-112;

- The research objectives remain unclear;

- The selection of variables included in the models is not clearly explained, especially the non-integration of land use and land cover and the integration, for example, of powerlines.  Clarify, for example, the buffer around the voltage transport network.

The data presented in table 2 should be clarified and should be congruent with the maps that follow.

Also in the Table 2 clarifies what "fire status" represents. The figures don't seem to tally with the text previously mentioned for the number of fires and the area burnt.

The legends on maps 9 to 14 need to be revised, as there is no continuity in terms of classes.

The values assigned to the classes presented for powerlines seem excessive, and their importance in the occurrence of fires is unclear.

In the text (lines 492-93), the authors state: “The RF and XGB models have indicated the highest importance for the population factor” which is not correct according to the figure 21 e 22. In fact, it's very strange that the most important factor was the powerlines, a result that is not discussed in the light of the bibliography.

The conclusion should be revised, focussing on the main results and limitations of the work presented.

I think that the title of the work should be reflected since it is about the risk of ignition/occurrence of fire and not the risk of forest fire (which should include ignition and burnt area). It should also reflect on the variables that are usually included in analyzing the risk of fire ignition, as they are very different from the risk of fires, which is mostly related to the area burnt.

Author Response

Dear Reviewer,

I sincerely appreciate the time you reviewing our manuscript and reccommending suggestions for do it acceptible for the journal.  We hope our revised manuscript is improved enought.  

Best regards

Reviewer 2 Report (New Reviewer)

Overall:

This is an interesting paper.  It describes a problem with wildfire in the WUI and identifies techniques to quantify risk using GIS.  

Abstract line 17 This part is not necessary "across different parts of the world" edit

lines 21-23 to read more fluidly.  I suggest: The Land use/land cover (LULC) characteristics of the region were examined and machine learning techniques including Random Forest (RF), Extreme Gradient Boosting (XGB), and Light Gradient Boosting (LGB) models applied to classify factors affecting fires.  

Introduction Line 36: it is not a good idea to state that anything is a "known fact".  Furthermore, the authors did not support this claim with even one citation.  This must be improved and corrected.    

Again, a citation is needed on line 45. "...clearly will contribute..."   The authors are omitting an important consideration and viewing this problem in short-term only.  It reads that fires must be extinguished immediately or as soon as possible.  That reads much like the old 10am policy of the US Forest Service.  Ultimately, that led to forests with tremendous ladder fuels, fuel stockpiling, and the conflagrations seen today.  This concept needs to be addressed by the authors.  

Methods Figure 1 looks very good.  Please edit though, to remove the "Turkey_Istanbul_Europa" text and improve the caption.  Understand that all figures and tables (along with their captions) must be able to stand on their own.  Be sure to write excellent captions for all figures and tables.  

Line 177, it would be informative to see what the population density is (XXX/km2) rather than read it "is densely populated".  

Table 2, what is COUNT?  Is this from raster data and it is simply a count of pixels?  If so, it should be deleted.  Again, a much better caption is required.  

Figure 4, there appears to be a black square in the upper right of this figure. I have no clue what this is.  

Is figure 9 actually a DSM or DTM?  

Line 317: Proximity to residential areas is considered risky due. Do the authors mean "Proximity to residential areas increases risk"???  

Table 5, the equation listed to the left of (5) is not fully readable. ?1=2  (it looks like the line number 435 printed over it).  I hope this can be fixed.   Distance to powerlines is quite coarse.  If a fire starts due to a powerline, it will typically very close (<100 meters).  Did the authors use a classified or categorical distance layer as shown in figure 14 or were the actual data continuous?  

Results/Discussion The results are interesting and are supported by the data and methods described.   I wonder is the population factor selected by RF the population density layer?  Or were other population metrics used.  Clarify this and use the same terminology throughout the paper.  

Conclusions   The most important test of accuracy or correctness is the reality test.  The authors noted that proximity to water was selected as the most important factor by the LGB classifier.  It would be beneficial if the authors would interpret these models. 

Why would DW be most important? 

How was it used? Was it inversely applied?

Did fire decrease as DW decreased? 

This could be very real as vegetation near water sources (rivers) would tend to be wet or even wetlands vegetation.

There are a few places where English can be improved but overall it is well written and easily understood.

Author Response

Dear Reviewer,

I sincerely appreciate the time you reviewing our manuscript and reccommending suggestions for do it acceptible for the journal.  We hope our revised manuscript is improved enought.  

Best regards

Round 2

Reviewer 1 Report (Previous Reviewer 1)

Presenting a normalised legend for the maps doesn't seem appropriate for characterising the study area, as it takes away from the spatial reading of the variables used. Instead of classes, you could use maximum and minimum values.

Caption the acronyms of the variables shown in figure 21, as they can lead to confusion. Does PO stand for population density or power lines?

Author Response

Dear Reviewer, 
 Answers to your comments and suggestions are presented below. 
Best regards

Comments and Suggestions for Authors

Presenting a normalised legend for the maps doesn't seem appropriate for characterising the study area, as it takes away from the spatial reading of the variables used. Instead of classes, you could use maximum and minimum values.

The minimum and maximum values of the layers are given in all labels in the maps as proposed. Additionally, the number those expressed the highest slope which have written by mistakenly, have been changed with the correct value. The following next sentence was deleted as it was supporting the old previous incorrect value of slope (line 291).

Caption the acronyms of the variables shown in figure 21, as they can lead to confusion. Does PO stand for population density or power lines?

The acronyms on the graph in Figure 21 have been changed with their full names.

This manuscript is a resubmission of an earlier submission. The following is a list of the peer review reports and author responses from that submission.

Round 1

Reviewer 1 Report

The title of the article is very suggestive, but its development needs to be substantially improved for the following reasons:

- The objectives are not well defined.

- The introduction does not present a guiding line about the concept of risk and the variables and methodologies used for its determination; it does not discuss the central theme of the study: Fire Risk in Wildland-Rural and Urban Interface Regions

- In the introduction, substantial information is irrelevant for the subject to be discussed (for example between lines 60-70 or between lines 109-131).

-In the materials and methods, the study area needs a better characterization; there is a lack of justification of the variables selected for the fire risk analysis; it is surprising that the land use/land cover is not among the selected variables…

For example, what is the meaning of Figure 12. Population? or Figure 14. Fire and relationship maps.

The methodologies used in the risk calculation need to be better detailed.

In the results, where are the maps with the Fire Risk Assessment in Wildland-Rural and Urban Interface Regions?

I cannot understand this conclusion: The model used was successful because the factors that are effective in the start of  the fire and the factors that are effective in its spread can be realistically separated with  the parameters selected regarding the region where the study was conducted.

Reviewer 2 Report

The manuscript discusses the application of machine learning methods for assessing fire risk in Wildland-Rural and Urban Interface regions on the example of Istanbuls European side. The work considers random forest, extreme gradient boosting, and light gradient boosting. The evolution of these methods shows good performance of random forest approach in comparison with other methods. The advantage of the presented method is the use of publicly available data, which simplifies the application to other study areas.

Even the manuscript is devoted to the evaluate the machine learning techniques for fire risk assessment in Wildland-Rural and Urban Interface regions, the methodology section presents machine learning techniques very briefly and the authors did not bring any innovations to the configuration of these methods. Comparative analysis in this case is limited since evaluating the machine learning techniques are very dependent on input data and with a variability of the characteristics associated with fires, the result of the effectiveness of the methods may change (that was described in the end of introduction).

However, the work is of great interest due to outcomes of features importance analysis. As expected, based on the introduction of the manuscript, proximity to settlements and high population density is a significant fire risk factor as it increases anthropogenic pressure. The novelty of the work is the detection of powerlines as an important fire risk factor in Wildland-Rural and Urban Interface regions. This outcome should be covered in more detail in the manuscript, as it is non-trivial in the system of risk assessment and fire management and can have a great impact on fire risk assessment approach in Wildland-Urban Interface.

Specific comments

Lines 60-62: Add reference to the sentence

Table 1 repeats Table 2. Recommended to put Table 2 instead Table 1

Lines 103-147: Describes study area, recommended to move to the methodology section

Line 149: GIS is not a method. It is the system (platform) for store, manage, analyze, edit, output, and visualize data

Line 163: You mention good performance of ANN with radial basis function. Why dont you test it in the study?

Line 174: At the end of the introduction, there is no clearly formulated aim of the study.

Section 2.3  (line 364) the sequence of sections confuses: K-fold cross-validation presented in the same format as machine learning methods (looks like 4th method instead of methodological description of data splitting for training)

Figures 20-22: I suggest using full name of features in the graph

Lines 512-514: The sentence is not clear. The manuscript doesnt discuss the separation of factors that are effective in the start of the fire and the factors that are effective in its spread

Figures 20-22: Change in the figure description form future to feature